# The Long-Term Follow-Up of Patients with Cystine Stones: A Single-Center Experience for 13 Years

**DOI:** 10.3390/jcm10071336

**Published:** 2021-03-24

**Authors:** Toshifumi Takahashi, Shinya Somiya, Katsuhiro Ito, Toru Kanno, Yoshihito Higashi, Hitoshi Yamada

**Affiliations:** Department of Urology, Ijinkai Takeda General Hospital, 28-1 Moriminami-cho, Ishida Fushimi-ku, Kyoto 601-1495, Japan; t.takahashi201@gmail.com (T.T.); ssomiya@kuhp.kyoto-u.ac.jp (S.S.); itokatsu@kuhp.kyoto-u.ac.jp (K.I.); t.kan@kuhp.kyoto-u.ac.jp (T.K.); yyyhigashi@kyoto.med.or.jp (Y.H.)

**Keywords:** cystine stones, clinical characteristics, clinical course, urolithiasis, treatment outcome

## Abstract

Introduction: Cystine stone development is relatively uncommon among patients with urolithiasis, and most studies have reported only on small sample sizes and short follow-up periods. We evaluated clinical courses and treatment outcomes of patients with cystine stones with long-term follow-up at our center. Methods: We retrospectively analyzed 22 patients diagnosed with cystine stones between January 1989 and May 2019. Results: The median follow-up was 160 (range 6–340) months, and the median patient age at diagnosis was 46 (range 12–82) years. All patients underwent surgical interventions at the first visit (4 extracorporeal shockwave lithotripsy, 5 ureteroscopy, and 13 percutaneous nephrolithotripsy). The median number of stone events and surgical interventions per year was 0.45 (range 0–2.6) and 0.19 (range 0–1.3) after initial surgical intervention. The median time to stone events and surgical intervention was 2 years and 3.25 years, respectively. There was a significant difference in time to stone events and second surgical intervention when patients were divided at 50 years of age at diagnosis (*p* = 0.02, 0.04, respectively). Conclusions: Only age at a diagnosis under 50 was significantly associated with recurrent stone events and intervention. Adequate follow-up and treatment are needed to manage patients with cystine stones safely.

## 1. Introduction

Cystine stone development is relatively uncommon among urolithiasis patients, about 1% in adults and 6–10% in children [1,2]. Cystine stone formers have a genetic disorder in which cystine is absorbed in the proximal tubule. Since cystine solubility in urine is very low, patients who form cystine stones often suffer from stone recurrences and regrowth and require surgical interventions [3]. Cystinuria is diagnosed by family history, stone analysis, or microscopic examination of the urine and 24-h urine testing. Managing patients with cystine stones is challenging because of the lifelong risk of stone recurrence. Current preventive treatments for cystinuria include increased fluid intake to increase cystine solubility and limiting sodium and protein intake to decrease cystine excretion. Pharmacologic therapy such as alkalizing agents and cystine-binding thiol drugs may also be effective [4]. Although it has been reported that patients taking cystine-binding thiol drugs have a high quality of life, there are adverse effects and poor compliance with the drugs [5,6]. Despite these preventive measures, cystine stone formers frequently experience stone-related episodes and surgical intervention [4]. Many patients suffer from renal insufficiency because of the stone-recurrence. Thus, life-long follow-up is necessary to monitor kidney function, find recurrences at an early stage, and increase patient compliance of dietary and medical treatments [4]. However, because of the rarity of the disease, the appropriate treatment and follow-up strategy to reduce the risk of stone events and avoid loss of renal function has not been elucidated.

Since this disease is so rare, most studies have reported only on small sample sizes (10–30 patients) and short follow-up periods (5–10 years) [3,7,8,9,10,11,12,13,14,15,16]. Herein, we report our single-center experience of 22 cases of patients with cystine stones. The data of these patients were gathered for a median of 13 years and will help us to further understand the characteristics of these patients. We also examined the risk factors of repeat intervention.

## 2. Material and Methods

### 2.1. Patients

The study protocol was reviewed and approved by the Institutional Review Board at Ijinkai Takeda General Hospital (IRB no. 20200002). We retrospectively analyzed 22 patients with cystine stones diagnosed between January 1989 and May 2019, and with follow-up periods of >6 months. For all patients, a diagnosis of cystine stone was confirmed by routine laboratory stone analysis of ex vivo stone specimens obtained after stone passage or stone extraction procedures for all patients. Patient age at diagnosis, sex, number, location, laterality and diameter of stones, maximum stone size after initial treatment, use of alkalinizing agents or thiol-binding agents, stone events, surgical treatments, and estimated glomerular filtration rate (eGFR) were the variables reviewed. Stone events were defined as renal colic, stone growth, asymptomatic ureteral stone, and surgical intervention. Stone growth was defined as an increase greater than 50% in the diameter of stones larger than 5 mm. Surgical interventions were percutaneous nephrolithotomy (PCNL), ureteroscopy (URS), and extracorporeal shock wave lithotripsy (ESWL). Auxiliary treatments for the same stone or residual fragment were not counted as events. The number, location, and diameter of stones were examined by ultrasonography (US).

### 2.2. Follow-Up Protocol

After the initial treatment, we evaluated each patient via US about 1 month after discharge. All patients were given instructions regarding daily fluid intake, low-sodium diet, and low-protein diet, and had a follow-up urinalysis, including urine specific gravity and urine pH. Alkalinizing agents (6 g of potassium citrate or 3 g of sodium bicarbonate) were administered to achieve urine pH > 7.5. A thiol-binding agent (600 mg of tiopronin) was prescribed depending on frequency of stone-related events. Patients were required to receive US screening every 3–6 months. We performed computed tomography (CT) scans and plain abdominal radiography when symptoms suggested ureteral obstruction because of the risk of radiation exposure. Based on Japanese guideline on urolithiasis [17], in the presence of renal stone growth or renal colic, renal stones of 5–20 mm in size were generally treated with ESWL. The flexible URS (fURS) became available in our institution in 2012. After that, lower pole renal stones < 20 mm or ESWL-resistant stones were treated with fURS. A renal stone larger than 20 mm was usually treated with PCNL. Patients were treated with URS for ureteral stones larger than 10 mm or stones that did not spontaneously pass within one month. When performing ESWL, we used ultrasonography for localization of renal stones, whereas fluoroscopy with or without intravenous urography was utilized for ureteral stones. We used a flexible ureteroscope (URF-V3, 8.4 F. or URF-V, 9.9Fr, Olympus, Tokyo, Japan) and semi-rigid ureteroscope (8.0/9.8Fr, Richard Wolf, Knittlingen, Germany) in URS and a nephroscope (26/24Fr, Richard Wolf, Germany) in PCNL. The double-J stent was inserted in the case of obstructive pyelonephritis or ureteral/renal calculi for which URS was planned (pre-stenting), and the stent placement was not used in the cases when ESWL was performed.

### 2.3. Statistical Analysis

All statistical analyses were performed with EZR [18]. Wilcoxon signed-rank test and Pearson chi-square test were used to check for statistical significance where appropriate. The Kaplan–Meier method was used to estimate the probability of stone events and surgical interventions stratified by patient age at diagnosis, sex, staghorn calculi or others, use of medications (alkalinizing agents or thiol-binding agent) after the initial treatment, and postoperative stone size (<5 mm or ≥5 mm). The cut-off of the age (under 50 or 50 and over) was determined by previous reports [19,20] The number of stone events or surgical interventions per year was calculated as the number of stone events or surgical interventions divided by the number of years of observation. The cumulative incidence rate was calculated by Kaplan–Meier estimates. The log-rank test was used to compare the rate of stone events and surgical interventions in these variables.

## 3. Results

Twenty-two patients underwent a median follow-up of 13.3 years (range 0.5–28.3). The median patient age at diagnosis was 46 (range 12–82). Fourteen were male, and eight were female. At the initial presentation, 16 patients had stones in bilateral sides. Seventeen patients had only renal stones, whereas 3 patients had ureteral stones, and 2 patients had both renal and ureteral stones. Staghorn calculi were present in 14 patients. All patients underwent surgical interventions during their first visit. In the stone analysis, 20 patients had pure cystine stones and 2 patients had cystine-containing stones. The procedures included 4 ESWL, 5 URS, and 13 PCNL. Thirteen patients with staghorn calculi underwent PCNL; 1 patient with staghorn calculi, 1 patient with ureteral stones, 1 patient with renal stones, and 2 patients with both renal and ureteral stones underwent URS (auxiliary ESWL was added at a physician’s discretion); and 2 patients with renal stones and 2 patients with ureteral stones underwent ESWL. The median stone diameter before the first surgical intervention was 30 (range 5–66) mm, and that after the first surgical intervention was 5 (range 3–9) mm. After diagnosis, sixteen patients took medical therapy, 3 were taking potassium citrate, 1 tiopronin, 1 sodium bicarbonate, 8 potassium citrate and tiopronin, 1 tiopronin and sodium bicarbonate, and 2 were taking potassium citrate, tiopronin, and sodium bicarbonate (Table 1).

Figure 1 presents the stone events and surgical intervention of all patients. The median number of stone events per year was 0.45 (range 0–2.6), and the median number of surgical interventions per year was 0.19 (range 0–1.3) after initial surgical intervention. The median stone size at the secondary and subsequent surgical interventions was 8 mm (range 5–18). Two patients had a fever before surgical intervention, and one of them had obstructive pyelonephritis and underwent stent placement. There were 138 surgical interventions. The indications for surgery were stone growth (77 (55.8%)), renal colic (42 (30.4%)), asymptomatic ureteral stone (6 (4.3%)), and patients’ wishes (13 (9.4%)). Surgical interventions included 122 (88.4%) ESWL, 13 (9.4%) URS (±ESWL), and 3 (2.2%) PCNL (±ESWL) for 17 patients (Table 2). Because most ESWL procedures in this study were intended to reduce future symptomatic stone-related events, we performed a separate analysis after exclusion of stone growth and ESWL. The median number of symptomatic stone events and surgical intervention per year except for stone growth and ESWL was 0.09 (range 0–0.95) and 0.0 (range 0–0.34), respectively. Eight patients underwent fURS during follow-up, and the median number of symptomatic stone events and surgical intervention per year after fURS was 0.0 (range 0–1.14) and 0.0 (range 0–0.43), respectively.

Kaplan–Meier curves of time to any stone event or the second surgical intervention for all patients are shown in Figure 2A,B. The median time to stone events and surgical intervention was 2 years and 3.25 years, respectively. After stone growth and ESWL were excluded, the median time to symptomatic stone events and surgical interventions per year were 5.59 and 19.85 year, respectively. The median time to any stone event or the second surgical intervention stratified by patient characteristics and stone-related factors are shown in Table 3. There was a significant difference in time to stone events and second surgical intervention when patients were analyzed using a cutoff of 50 years of age at diagnosis (1.50 vs. 5.00 years, *p* = 0.02, and 1.50 vs. 5.00 years, *p* = 0.04, respectively). There were no significant differences when patients were analyzed by sex, staghorn calculi or not, medication, and postoperative stone size (<5 mm or not). When stone growth and ESWL were excluded, age at diagnosis was still a significant factor for stone event (2.00 vs. 11.4 years, *p* = 0.008) but not for surgical intervention (18.59 vs. not reached, *p* = 0.25). Kaplan–Meier curves of time to any stone event or the second surgical intervention stratified by patient age at diagnosis are shown in Figure 3A,B.

Renal function was estimated several times during follow-up for 18 patients who had available data. The first measured median eGFR was 57.0 (range 35.0–92.1) mL/min/1.73 m^2^, and the last measured median eGFR was 58.2 (range 25.7–75.1) mL/min/1.73 m^2^. There was no significant correlation between the change of eGFR and the total treatment frequency per patient (r^2^ = −0.24, *p* = 0.34) (Figure 4).

## 4. Discussion

In this study, we comprehensively reviewed the clinical progress of patients with cystine stones. There was a significant difference in time to stone events and second surgical intervention only when patients were analyzed using a cutoff of 50 years of age at diagnosis among various risks. There was no clear correlation between the decline in the renal function and the number of surgical interventions.

Herein, patients frequently presented stone recurrences. Previous studies have reported a high frequency of repeat surgeries, and the overall mean number of surgical procedures per patient-year was 0.22–1.32 [3,7,8,9,10,11,12,13,14,15,16,21]. Our study showed that the number of surgical procedures per patient-year was 0.24, and the median interval between repeat surgeries was 3.25 years, which is consistent with previous reports. PCNL was more frequently utilized as the first surgical treatment due to the initial large stone size, while ESWL was performed more frequently after that. Cystine stones are considered resistant to ESWL. However, cystine stones with heterogeneous, “rough”, morphology has been recognized as ESWL-susceptible compared to those with “smooth” morphology [22,23]. Moreover, cystine stones in the early course of a recurrence may tend to show rough morphology [24]. Of note, we actively treated asymptomatic growing renal stones with ESWL, resulting in a higher proportion of ESWL use. After excluding stone growth and ESWL, the incidence of symptomatic stone events and surgical intervention was lower than that seen in other studies. This result suggested a prophylactic role of ESWL for asymptomatic renal stones. It has been reported that the CT features of the stones may be helpful for candidate selection of ESWL [22,23]. Moreover, tiopronin has been suggested to make cystine stones more fragile [25,26]. Thus, early detection of “rough” cystine stones during regular follow-up and active stone fragmentation via ESWL with CT and tiopronin could be a new approach to manage patients. Recently, URS has increasingly been used for the treatment of renal stones. Compared to ESWL, URS have an advantage in clearance rate of residual fragments, particularly for the lower pole stones [27]. Actually, patients who underwent fURS in this study had fewer subsequent stone events and surgical interventions after that. The best timing and modality to treat growing renal stones should be further investigated. The incidence of stone events and surgical intervention varied widely among patients. This difference in the recurrence rate indicates the need for an individualized follow-up period and intervention.

In this study, an age at diagnosis less than 50 years was significantly associated with repeat stone events and surgical interventions. The association between younger age and risk of repeat events in patients without cystinuria has been reported [28]. It may reflect more severe metabolic abnormalities leading to stone formation at younger ages. Although chromosomal analysis was not performed in this study, the variation of genotypes may influence the difference. Recently proposed classifications of cystinuria include Types A and B [29]. In Type B cystinuria, heterozygotes, and homozygotes with the gene mutation responsible for cystinuria present increased urinary cystine secretion. A significant difference in age at the initial stone episode was found between heterozygotes and homozygotes [19]. It was reported that patients with cystinuria with a first stone event at an older age had lower cystine concentration in urine [20]. In this regard, this study is the first study to show that age at diagnosis is also associated with repeat stone events and surgical interventions. The association should be further investigated with the chromosomal analysis and 24 h urine test.

The presence of staghorn calculi at the initial visit was not associated with repeat stone events or surgical intervention, cystine stones are often found as staghorn calculi at some patients’ first visit. Rhodes et al. reported that 20% of patients with cystine stones had staghorn calculi [30]. In our report, 14 patients (64%) were diagnosed with staghorn calculi at their first visit. Many patients with staghorn calculi are referred to our high-volume endourology centers, which account for the relatively high incidence. However, no reports have stated whether the presence of staghorn calculi at the initial diagnosis is a risk factor for stone recurrence. Interestingly, patients with staghorn calculi at initial presentation had lower risk of future events than ones without staghorn calculi. This result indicates that not the initial stone size but other factors such as age or treatment success may have a greater impact on recurrent events. 

In this study, we observed longer no-stone event periods and intervention-free survival in patients with postoperative fragment <5 mm, although the difference was not significant due to the small sample size. Moore et al. reported the importance of being entirely stone-free and fragment-free after their first intervention [3]. The report showed that the recurrence rate was 85.7% in patients who were not stone-free after their first treatment, while the recurrence rate was as low as 44.4% among patients who were found to be stone-free [3]. Even small residual fragments (<2 mm) were reported to be associated with repeat intervention in patients with non-cystine stones [31]. The optimal cut-off value of residual fragment should be further examined.

Avoiding loss of renal function is fundamental for patients with cystine stones. Due to multiple stone recurrences, patients with cystine stones tend to have lower renal function compared to patients that develop other types of stones [32]. Notably, the renal function of patients in our study was almost preserved and was not associated with the number of surgical interventions. In contrast, Moore et al. reported that renal function might decline in patients who undergo more than four surgical interventions [3]. In our report, more than half of the indications for surgical intervention were related to stone growth, and most of the procedures were ESWL and URS. The high proportion of ESWL is likely related to lack of a flexible ureterorenoscope, which became available in our institution in 2012. Although the long-term effect of URS or ESWL is under debate, these modalities are reported to have few effects on renal function [33,34,35]. Moreover, our routine follow-up found six cases of asymptomatic ureteral stones. Silent ureteral stones considerably increase the risk of irreversible renal function impairment [36]. Active treatment for stone growth and close monitoring every 3–6 months are possible reasons why the renal function of our patients was successfully preserved.

The present study has some limitations that should be acknowledged. First, because of the characteristic of our institution, this study included patients who were incidentally diagnosed at the time of surgery; patients diagnosed at pediatric screening were not included. Thus, symptoms before the initial intervention and changes in the rate of stone-related complication after intervention remain unknown. Second, this study is retrospective, and some of the clinical data, such as renal function or stone events without intervention, are not complete. Third, although all patients in this study received the guidance of diet and fluid intake to maintain appropriate urine pH, its compliance was unknown. Fourth, stones are measured by US, which may be inaccurate compared with CT measurements. However, the use of CT should be minimal for patients with cystine stones because there is a non-negligible risk of ionizing radiation due to lifelong follow-up. Fifth, patients with cystine stones were diagnosed based on the stone analysis. Because cystine excretion tests and genetic analyses were not performed, evaluation such as urine cystine levels and urinary cystine supersaturation could not be performed. Sixth, we could not perform multivariate analysis because of the small number of patients. To address the abovementioned limitations, a large-scale, multicenter prospective study should be conducted for further understanding of this rare disease. Despite these limitations, this is one of the most extended studies to report the characteristics of patients with cystine stones in a single institution. Further, patients were observed safely without any significant renal dysfunction. This study helps confirm the natural history and clinical outcomes of cystine stone disease.

## 5. Conclusions

The long-term observation of 22 patients with cystine stones highlighted a wide variety of frequency in stone events and needed surgical interventions. Only age at diagnosis under 50 years was significantly associated with recurrent stone events and surgical interventions. The patients with cystine stones were safely managed without loss of renal function thanks to close follow-up and active treatment. This study suggested the potential of ESWL for a prophylactic role. Further large-scale research is needed to individualize follow-up treatment schedules.

## Figures and Tables

**Figure 1 jcm-10-01336-f001:**
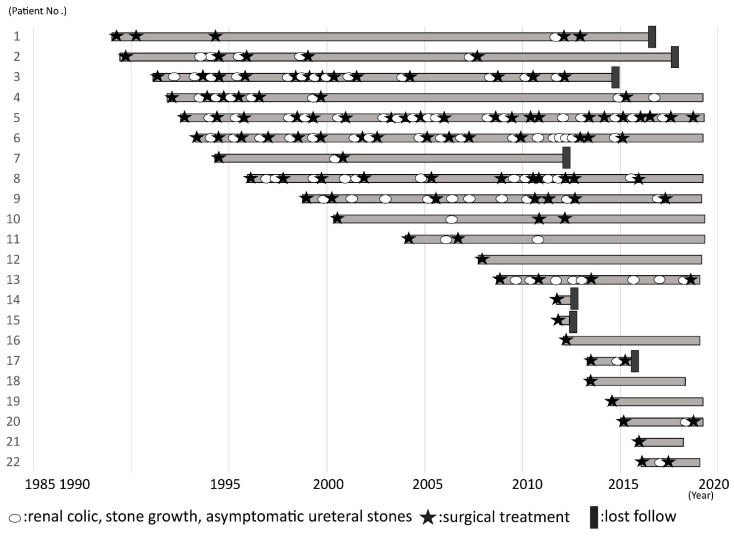
Patients’ clinical course after first treatment.

**Figure 2 jcm-10-01336-f002:**
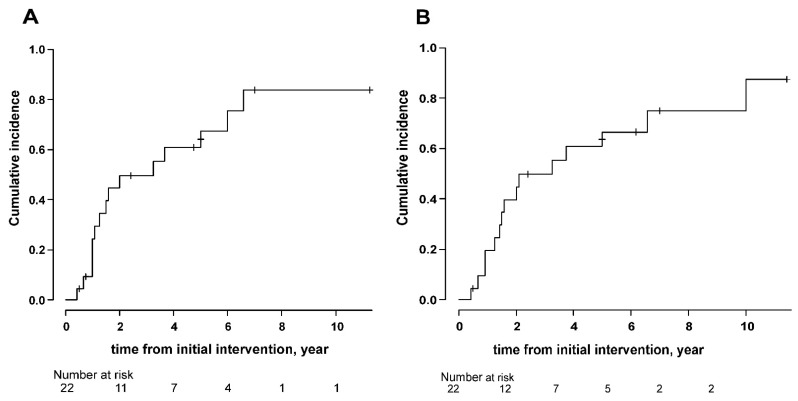
Cumulative incidence of stone events (**A**) and second surgical intervention (**B**) of all patients.

**Figure 3 jcm-10-01336-f003:**
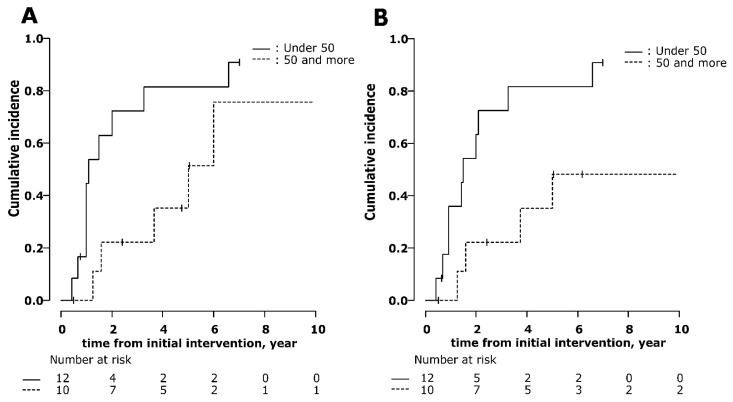
Cumulative incidence of stone events (**A**) and second surgical intervention (**B**) of patients under age 50 and 50 and over at diagnosis.

**Figure 4 jcm-10-01336-f004:**
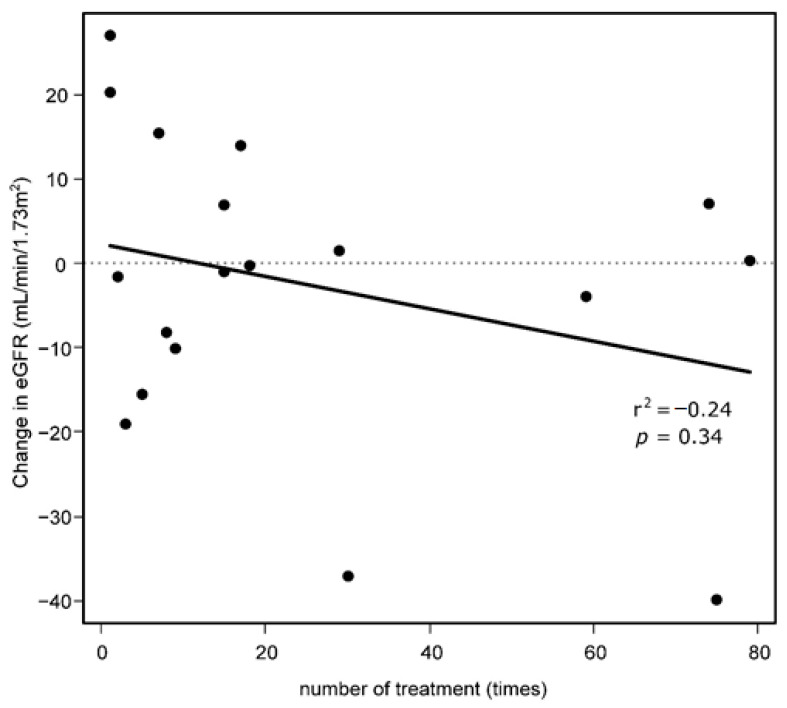
Correlation between change of estimated glomerular filtration rate (eGFR) and treatment frequency.

**Table 1 jcm-10-01336-t001:** Patient characteristics.

Variable	Value
Age at diagnosis, year	46 (12–82)
Sex (male/female)	14:8
Duration of follow-up, month	160 (6–340)
Stone formation, unilateral/bilateral	6:16
Stone location, renal/ureter/both	17 (staghorn calculi 14): 3:2
Surgical interventions at first visit	
ESWL	4 (18)
URS	5 (22)
PCNL	13 (59)
Stone size	
Before the first surgical intervention, mm	30 (5–66)
After the first surgical intervention, mm	5 (3–9)
Stone components (pure cystine/mixed)	20:2 (mixed with calcium phosphate)
Medications	
Potassium citrates	13 (59)
Tiopronin	12 (55)
Sodium bicarbonates	4 (18)
No medication	6 (27)

Results are shown as the median (range), the number, or the number (%). ESWL, extracorporeal shockwave lithotripsy; URS, ureteroscopy; PCNL, percutaneous nephrolithotomy.

**Table 2 jcm-10-01336-t002:** Summary of treatment outcomes.

Variable	Value
Surgical interventions after first treatment	
ESWL	122 (84.4)
URS	13 (9.4)
PNL	3 (2.2)
The reasons for surgical interventions	
Stone growth	77 (55.8)
Pain	42 (30.4)
Asymptomatic ureteral stone	6 (4.3)
Patients’ wishes	13 (9.4)
Stone events per year	0.45 (0–2.6)
Surgical interventions per year	0.19 (0–1.3)
Stone events per year (excluding stone growth and ESWL)	0.09 (0–0.95)
Surgical interventions per year (excluding ESWL)	0 (0–0.34)
Stone events per year after fURS	0 (0–1.14)
Surgical interventions per year after fURS	0 (0–0.43)

Results are shown as the median (range) or the number (%); ESWL, extracorporeal shock wave lithotripsy; fURS, flexible ureteroscopy; PCNL, percutaneous nephrolithotomy.

**Table 3 jcm-10-01336-t003:** The time to any stone events or the second surgical interventions stratified by various risk factors as determined by Kaplan–Meier estimates.

		Median Time to Any Stone Events (Years)		Median Time to Surgical Intervention (Years)	
Risk Factors	*n*	All	Excluding Stone Growth and ESWL	All	Excluding ESWL
Age at diagnosis					
Under 50	12	1.50	2.00	1.50	18.59
50 and over	10	5.00	11.4	5.00	NR
		(*p* = 0.02 *)	(*p* = 0.008 *)	(*p* = 0.04 *)	(*p* = 0.25)
Sex					
Male	14	1.58	5.59	2.00	18.59
Female	8	3.67	5.00	3.75	21.43
		(*p* = 0.84)	(*p* = 0.52)	(*p* = 0.82)	(*p* = 0.45)
Staghorn calculi					
Presence	13	3.67	11.43	3.75	18.59
Absence	9	1.25	5.00	1.42	21.43
		(*p* = 0.19)	(*p* = 0.53)	(*p* = 0.30)	(*p* = 0.28)
Medications					
Presence	16	1.58	3.25	2.10	11.42
Absence	6	2.00	5.59	2.00	19.8
		(*p* = 0.39)	(*p* = 0.13)	(*p* = 0.34)	(*p* = 0.17)
Postoperative stone size					
<5 mm	13	5.00	11.43	5.00	21.4
≥5 mm	9	2.00	3.25	2.00	18.6
		(*p* = 0.62)	(*p* = 0.55)	(*p* = 0.29)	(*p* = 0.45)

Medications include alkalinizing agents (potassium citrate or sodium bicarbonate) and a thiol-binding agent (tiopronin). NR, not reached; ESWL, extracorporeal shockwave lithotripsy. * *p* < 0.05.

## Data Availability

Data sharing not applicable.

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
