# Peer review of "The Long-Term Follow-Up of Patients with Cystine Stones: A Single-Center Experience for 13 Years"

_jcm, 2021, doi:10.3390/jcm10071336_

Round 1
Reviewer 1 Report
Authors present quite interesting study about a rare cause of kidney stones in adults, which also is hard to control in clinical practice. As Authors mentioned, available studies analyzed small groups of patients , so in this paper quite big group was examined. Interestingly, the influence of age on patients stone-risk reocurrence was shown. was However if possible:
1) please use the full name of the procedure: Extracorporeal shock wave lithotripsy (ESWL)
2) do you have any data what was the time from diagnosis till stone occurrence? or urinary tract stones was first manifestation of cystinuria and patients need surgical treatment from the very beginning (figure 1)?
3) Did any patient have struvite stones (stone component: mixed in Table 1)?
4) Did patients have infectious complications before/after procedures? Did the rate of other complications decreased after surgeries?
5) if possible please try to explain better why PCNL was more often used at first stone episode, whereas ESWL later? is it because lower size of stones when symptoms reoccurred?
6) if patient had 3 procedures (PCNL, URS, SWL) You calculated this as different procedure, compared to patient with SWL only (Table 1&2)? Is it not better to calculate each procedure separately?
Author Response
Reviewer #1:
Authors present quite interesting study about a rare cause of kidney stones in adults, which also is hard to control in clinical practice. As Authors mentioned, available studies analyzed small groups of patients , so in this paper quite big group was examined. Interestingly, the influence of age on patients stone-risk reocurrence was shown. was However if possible:
Thank you for your encouraging comment on our study. We agree with your suggestions and have incorporated our responses into the R1 version (Blue color).
- please use the full name of the procedure: Extracorporeal shock wave lithotripsy (ESWL)
A1: Thank you for your advice. We will correct them respectively.
- do you have any data what was the time from diagnosis till stone occurrence? or urinary tract stones was first manifestation of cystinuria and patients need surgical treatment from the very beginning (figure 1)?
A2: Thank you for your comments. Due to the characteristics of our institution, all of the patients in this study were diagnosed at the time of surgical treatment; this study did not include patients diagnosed at pediatric screening. To clarify this point, we have revised the Figure 1. We also added the sentence to the limitation section as follows:
Page 9, line 52-54 in the revised manuscript;
First, because of the characteristic of our institution, this study included patients who were incidentally diagnosed at the time of surgery; patients diagnosed at pediatric screening were not included.
- Did any patient have struvite stones (stone component: mixed in Table 1)?
A3: Of the 22 patients, 2 had mixed cystine stones and calcium phosphate and none had struvite stones. We have added the data to Table 1 as follows;
Page 3, Table 1 in the revised manuscript:
|
Stone components (pure cystine: mixed) |
20: 2(mixed with calcium phosphate) |
- Did patients have infectious complications before/after procedures? Did the rate of other complications decreased after surgeries?
A4: Thank you for the question. No patients had infectious complication at the initial presentation. Two patients experienced urinary tract infection during follow-up period. Unfortunately, the rate of complications before the initial surgery was unknown in this study. We agree that it is an important limitation and acknowledged in the limitation section as follows:
Page 4, Line5-6 in the revised manuscript:
Two patients had a fever before surgical intervention while following up, and one of them had obstructive pyelonephritis and underwent stent placement.
Page 10, line 1-2 in the revised manuscript:
Thus, symptoms before the initial intervention and changes in the rate of stone-related complication after intervention remain unknown.
- if possible please try to explain better why PCNL was more often used at first stone episode, whereas ESWL later? is it because lower size of stones when symptoms reoccurred?
A5: Thank you for your comments. Because the initial stone size was relatively large (median 30mm), PCNL was more often used at first stone episode. Meanwhile, the stone size at symptoms recurrence was much smaller (median 8 (range 5–18) mm). Thus, ESWL was utilized more frequently. We have added the following sentences to the Result and Discussion section;
Page 4, line 4-5 in the revised manuscript
The median stone size at the secondary and subsequent surgical interventions was 8 mm (range 5–18).
Page 8, Line 10-12 in the revised manuscript:
PCNL was more frequently utilized as first surgical treatment due to the initial large stone size, while ESWL was performed more frequently after that.
- if patient had 3 procedures (PCNL, URS, SWL) You calculated this as different procedure, compared to patient with SWL only (Table 1&2)? Is it not better to calculate each procedure separately?
A6: When multiple procedures were performed for single stone, we regarded these procedures as one series of event. That is because we focused on not the number of procedures, but the frequency of cystine stone recurrence. To clarify this point, we have added the sentence to the method section as follows and revised Table 1, 2;
Page 2, line 21-22 in the revised manuscript:
Auxiliary treatments for the same stone or residual fragment were not counted as events.
We believe that incorporating your advice into the R1 version has improved the manuscript. Thank you once again.

Reviewer 2 Report
Thank you very for asking me to review this paper. The authors reported their case series of cystine stone patients who required treatments for their stone. The study is interesting since the disease is quite uncommon and data about cystine stone patients are always welcome. I have the following comments.
- Why stone-free patient was defined as “one with stones less than 5 mm by US”? Was just one stone or also more than one stone less than 5 mm each considered stone-free? This threshold could be a bias in the study results for three reasons:
- US has a lower detection rate compared to computed tomography (CT)
- A threshold of 5 mm is quite an old definition for stone-free and a lower threshold should be used. Moreover, Kanno et al recently demonstrated that about 20% of asymptomatic renal stones ≤5 mm require surgical treatment within 5 years (Kanno T, et al. The Natural History of Asymptomatic Renal Stones ≤5 mm: Comparison with ≥5 mm. J Endourol. 2020 Nov;34(11):1188-1194. doi: 10.1089/end.2020.0336). Then, 5 mm is not acceptable as a threshold to define a patient as stone-free.
- A 5 mm stone may be symptomatic.
This point should be better clarified.
- Please clarify whether (range) after the median value means interquartile range (IQR) or minimum and maximum and provide it in the text
- Figure 1 depicts the history of lithiasis in each patient and I found it interesting and useful. Are there any chances to have a better presentation of the clinical course in the graphic because it is not clear in some patients due to overlapping points in the bar chart?
- Cystine calculi are generally considered SWL resistant. It has long been recognized that cystine stones present in two morphologically different types (Kim SC, et al Cystine: helical computerized tomography characterization of rough and smooth calculi in vitro. J Urol. 2005;174(4 Pt 1):1468-70;. doi: 10.1097/01.ju.0000173636.19741.24. Bhatta KM, et al. Cystine calculi--rough and smooth: a new clinical distinction. J Urol. 1989 Oct;142(4):937-40. doi: 10.1016/s0022-5347(17)38946-2), and that stones with a “rough” morphology fragments more easily with shock waves than stones (Kim SC, et al. Cystine calculi: correlation of CT-visible structure, CT number, and stone morphology with fragmentation by shock wave lithotripsy. Urol Res. 2007 35(6):319-24. doi: 10.1007/s00240-007-0117-1) with “smooth” morphology
- Why so many SWL treatments were performed?
- Were the patients selected according to CT features?
- Did the use of SWL decrease with the introduction of flexible ureteroscopy? If yes, this should be better clarified.
Author Response
Thank you very for asking me to review this paper. The authors reported their case series of cystine stone patients who required treatments for their stone. The study is interesting since the disease is quite uncommon and data about cystine stone patients are always welcome. I have the following comments.
Thank you for your encouraging comment on our study. We agree with your suggestions and have incorporated our responses into the R1 version (Red color).
1)Why stone-free patient was defined as “one with stones less than 5 mm by US”? Was just one stone or also more than one stone less than 5 mm each considered stone-free? This threshold could be a bias in the study results for three reasons:
US has a lower detection rate compared to computed tomography (CT)
A threshold of 5 mm is quite an old definition for stone-free and a lower threshold should be used. Moreover, Kanno et al recently demonstrated that about 20% of asymptomatic renal stones ≤5 mm require surgical treatment within 5 years (Kanno T, et al. The Natural History of Asymptomatic Renal Stones ≤5 mm: Comparison with ≥5 mm. J Endourol. 2020 Nov;34(11):1188-1194. doi: 10.1089/end.2020.0336). Then, 5 mm is not acceptable as a threshold to define a patient as stone-free.
A 5 mm stone may be symptomatic.
This point should be better clarified.
A1: Thank you for the important comment. We agree that the threshold of 5 mm was inappropriate for stone-free in recent practice. Unfortunately, not all patients underwent CT scan after the first intervention, so we use US findings for postoperative evaluation. We also agree that the lower accuracy of US for small renal stones.
In this study, we set the cut-off value of residual stone as 5 mm because very few patients achieved more strict criteria (zero fragment or < 2mm). Thus, we have rephrased the word “stone-free” as follows;
Page 3, line 3 in the revised manuscript and other changed as appropriate.
postoperative stone size (< 5 mm or ≥ 5 mm)
Page 9, line 28-30 in the revised manuscript
we observed longer no stone events periods and intervention-free survival in patients with postoperative fragment < 5 mm, although the difference was not significant due to the small sample size.
2)Please clarify whether (range) after the median value means interquartile range (IQR) or minimum and maximum and provide it in the text
A2: Thank you for your advice. The results were shown as the median (range).
We have clarified this point as follows:
Footnotes in the Table 1 and 2 in the revised manuscript:
Results were shown as the median (range).
3)Figure 1 depicts the history of lithiasis in each patient and I found it interesting and useful. Are there any chances to have a better presentation of the clinical course in the graphic because it is not clear in some patients due to overlapping points in the bar chart?
A3: Thank you for your advice. As you say, the color tone was the same and it was hard to understand. Therefore, we changed the circle to white and the star to black.
4)Cystine calculi are generally considered SWL resistant. It has long been recognized that cystine stones present in two morphologically different types (Kim SC, et al Cystine: helical computerized tomography characterization of rough and smooth calculi in vitro. J Urol. 2005;174(4 Pt 1):1468-70;. doi: 10.1097/01.ju.0000173636.19741.24. Bhatta KM, et al. Cystine calculi--rough and smooth: a new clinical distinction. J Urol. 1989 Oct;142(4):937-40. doi: 10.1016/s0022-5347(17)38946-2), and that stones with a “rough” morphology fragments more easily with shock waves than stones (Kim SC, et al. Cystine calculi: correlation of CT-visible structure, CT number, and stone morphology with fragmentation by shock wave lithotripsy. Urol Res. 2007 35(6):319-24. doi: 10.1007/s00240-007-0117-1) with “smooth” morphology
Why so many SWL treatments were performed?
Were the patients selected according to CT features?
Did the use of SWL decrease with the introduction of flexible ureteroscopy? If yes, this should be better clarified.
A4: We agree that SWL for cystine stones generally have low success rate. Meanwhile, SWL is less invasive than other modalities to treat renal stones. We preferred SWL for small- to medium-sized kidney stones instead of PCNL or open surgery until the introduction of flexible ureterorenoscope in 2012 in our institution. We also agree that CT can be helpful to find the candidate of ESWL, though we did not actively utilize it.
Eight patients underwent flexible ureteroscopy during follow-up period and the use of SWL decrease after flexible ureteroscopy. According to the appropriate comment from the reviewer, we have added the median number of symptomatic stone events and surgical intervention per year after flexible ureteroscopy in the Result section and Table 2:
Page 4, line 14-17 in the revised manuscript:
Eight patients underwent fURS during follow-up, and the median number of symptomatic stone events and surgical intervention per year after fURS was 0.0(range 0–1.14) and 0.0 (range 0–0.43), respectively.
We have added the below sentence in the Discussion section.
Page 8, line 12-16 in the manuscript;
Cystine stones are considered resistant to ESWL. However, cystine stones with heterogeneous, “rough” morphology has been recognized as ESWL-susceptible compared to those with “smooth” morphology [22, 23]. Moreover, cystine stones in the early course of a recurrence may tend to show rough morphology [24].
Page 8, line 27-28 in the revised manuscript:
Actually, patients who underwent fURS in this study had fewer subsequent stone events and surgical interventions after that.
We believe that incorporating your advice into the R1 version has improved the manuscript. Thank you once again.

Reviewer 3 Report
Dear Authors,
Manuscript submitted by Takahashi et al. entitled “The long-term follow-up of cystine stone patients: a single-center experience for 13 years” described clinical courses and treatment outcomes of cystine stone patients in a single center. Researches for the evolution of cystine stone patients over such a long period of time are sparse, and the presented study provides a good source of information. However, the manuscript needs improvement.
Overall, the manuscript is well written. However the language
in some parts needs revising.
- Abstract - All patients who underwent surgical interventions included 4 shockwave lithotripsy, 5 ureteroscopy, and 13 percutaneous nephrolithotripsy at the first visit. - please reformulate; it is hard to understand.
- Introduction - "patients with urolithiasis" should be urolithiasis patients
- "the understanding of" should be understand
- Material and Methods - 2.2 "Because of the risk of radiation exposure, we performed computed tomography (CT) scan and plain abdominal radiography when symptoms suggested ureteral obstruction." should be We performed computed tomography (CT) scan and plain abdominal radiography when symptoms suggested ureteral obstruction because of the risk of radiation exposure.
- 2.3 [15,16].
- Results - "Sixteen patients took medical therapy after diagnosis, 13 were taking potassium citrate, 12 were taking tiopronin, and 4 were taking sodium bicarbonate" should be After diagnosis, sixteen patients took medical therapy, 13 were taking potassium citrate, 12 were taking tiopronin, and 4 were taking sodium bicarbonate
Please carefully read the whole manuscript and correct the English mistakes.
Please find below specific comments related to each section of the manuscript:
Introduction - needs to be revised; it is quite short. I suggest presenting the diagnosis modalities, management (medical and surgical), and follow-up for cystine stone patients.
Please include the following articles in the Introduction and Discussion section:
Moussa M, Papatsoris AG, Abou Chakra M, Moussa Y. Update on cystine stones: current and future concepts in treatment. Intractable Rare Dis Res. 2020;9(2):71-78. doi:10.5582/irdr.2020.03006
Prot-Bertoye C, Lebbah S, Daudon M, Tostivint I, Jais JP, Lillo-Le Louët A, Pontoizeau C, Cochat P, Bataille P, Bridoux F, Brignon P, Choquenet C, Combe C, Conort P, Decramer S, Doré B, Dussol B, Essig M, Frimat M, Gaunez N, Joly D, Le Toquin-Bernard S, Méjean A, Meria P, Morin D, N'Guyen HV, Normand M, Pietak M, Ronco P, Saussine C, Tsimaratos M, Friedlander G, Traxer O, Knebelmann B, Courbebaisse M; French Cystinuria Group. Adverse events associated with currently used medical treatments for cystinuria and treatment goals: results from a series of 442 patients in France. BJU Int. 2019 Nov;124(5):849-861. doi: 10.1111/bju.14721.
Modersitzki F, Goldfarb DS, Goldstein RL, Sur RL, Penniston KL. Assessment of health-related quality of life in patients with cystinuria on tiopronin therapy. Urolithiasis. 2020 Aug;48(4):313-320. doi: 10.1007/s00240-019-01174-6.
Material and Methods
- please define clearly the minimum period of follow-up for the patients included in the study
- 2.2 please specify if you use JJ or not, because you performed SWL in renal colic
- please specify according to which guidelines you decide when to perform SWL, ureteroscopy or PCNL
- When performing SWL, do you use sonography or fluoroscopy for localization of the stone?
- Please define exactly "a low likelihood of spontaneous passage."
Results
- please specify what you understand by bilateral stone, kidney, kidney and ureter...
- please reformulate "both stones"
- you mention 14 - staghorn calculi, but you performed 13 PCNL initially; 3 patients ureteral stones - 5 URS ???
- how do you evaluate the patient after first intervention?
- 16 patients - medical therapy (13/12/4) - what type of combination?
- Figure 3 - please try to improve the quality and text (bold text on X- and Y-axis)
Discussion
I think it is - In this study, we comprehensively presented
"This result suggested the prophylactic role of SWL for asymptomatic cystine renal stones" - is interesting. The authors should try to elaborate more on this aspect. This can be one of the major conclusions of the manuscript.
Author Response
Manuscript submitted by Takahashi et al. entitled “The long-term follow-up of cystine stone patients: a single-center experience for 13 years” described clinical courses and treatment outcomes of cystine stone patients in a single center. Researches for the evolution of cystine stone patients over such a long period of time are sparse, and the presented study provides a good source of information. However, the manuscript needs improvement.
Overall, the manuscript is well written. However the language in some parts needs revising.
Thank you for your encouraging comment on our study. We agree with your suggestions and have incorporated our responses into the R1 version (Green color).
1)Abstract - All patients who underwent surgical interventions included 4 shockwave lithotripsy, 5 ureteroscopy, and 13 percutaneous nephrolithotripsy at the first visit. - please reformulate; it is hard to understand.
A1: Thank you for your advice. We have rewritten the sentence as follows:
All patients underwent surgical interventions at the first visit (4 extracorporeal shockwave litho-tripsy, 5 ureteroscopy, and 13 percutaneous nephrolithotripsy).
2)Introduction - "patients with urolithiasis" should be urolithiasis patients
A2: Thank you for your advice. We have corrected the phrase.
3) "the understanding of" should be understand
A3: Thank you for your advice. We have corrected the phrase.
4)Material and Methods - 2.2 "Because of the risk of radiation exposure, we performed computed tomography (CT) scan and plain abdominal radiography when symptoms suggested ureteral obstruction." should be We performed computed tomography (CT) scan and plain abdominal radiography when symptoms suggested ureteral obstruction because of the risk of radiation exposure.
A4: Thank you for your advice. We have revised it as you pointed out.
5)Results - "Sixteen patients took medical therapy after diagnosis, 13 were taking potassium citrate, 12 were taking tiopronin, and 4 were taking sodium bicarbonate" should be After diagnosis, sixteen patients took medical therapy, 13 were taking potassium citrate, 12 were taking tiopronin, and 4 were taking sodium bicarbonate]
A5: Thank you for your advice. We have revised it as you pointed out.
6)Introduction - needs to be revised; it is quite short. I suggest presenting the diagnosis modalities, management (medical and surgical), and follow-up for cystine stone patients.
Please include the following articles in the Introduction and Discussion section:
Moussa M, Papatsoris AG, Abou Chakra M, Moussa Y. Update on cystine stones: current and future concepts in treatment. Intractable Rare Dis Res. 2020;9(2):71-78. doi:10.5582/irdr.2020.03006
Prot-Bertoye C, Lebbah S, Daudon M, Tostivint I, Jais JP, Lillo-Le Louët A, Pontoizeau C, Cochat P, Bataille P, Bridoux F, Brignon P, Choquenet C, Combe C, Conort P, Decramer S, Doré B, Dussol B, Essig M, Frimat M, Gaunez N, Joly D, Le Toquin-Bernard S, Méjean A, Meria P, Morin D, N'Guyen HV, Normand M, Pietak M, Ronco P, Saussine C, Tsimaratos M, Friedlander G, Traxer O, Knebelmann B, Courbebaisse M; French Cystinuria Group. Adverse events associated with currently used medical treatments for cystinuria and treatment goals: results from a series of 442 patients in France. BJU Int. 2019 Nov;124(5):849-861. doi: 10.1111/bju.14721.
Modersitzki F, Goldfarb DS, Goldstein RL, Sur RL, Penniston KL. Assessment of health-related quality of life in patients with cystinuria on tiopronin therapy. Urolithiasis. 2020 Aug;48(4):313-320. doi: 10.1007/s00240-019-01174-6.
A6: Thank you for your comments. Based on the presented paper, I have added it to the introduction.
Page 1, line 5-19 in the revised manuscript:
Cystinuria is diagnosed by family history, stone analysis, or microscopic examination of the urine and 24- hour urine testing. Managing cystine stone patients is challenging because of the lifelong risk of stone recurrence. Current preventive treatments for cystinuria include increased fluid intake to increase cystine solubility and limiting sodium and protein intake to decrease cystine excretion. Pharmacologic therapy such as alkalizing agents and cystine-binding thiol drugs may be also effective.[4] Although it has been reported that patients taking cystine-binding thiol drugs have a high quality of life, there are adverse effects and poor compliance with the drugs[5,6]. Despite these preventive measures, cystine stone formers frequently experience stone-related episode and surgical intervention [4]. Many patients suffer from renal insufficiency because of the stone-recurrence. Thus, life-long follow-up is necessary to monitor kidney function, find recurrence at early stage, and increase patient compliance of dietary and medical treatments [4]. However, because of the rarity of the disease, the appropriate treatment and follow-up strategy to reduce the risk of stone events and avoid loss of renal function has not been elucidated.
Material and Methods
- please define clearly the minimum period of follow-up for the patients included in the study
A7: Thank you for your comments. The study included patients who were followed for at least six months. We have added the minimum period of follow-up to the Method as follows:
Page 2 line 9-11, in the revised manuscript:
We retrospectively analyzed 22 patients with cystine stones diagnosed between January 1989 and May 2019, and with follow-up periods of > 6 months.
8)2.2 please specify if you use JJ or not, because you performed SWL in renal colic.
A8: Thank you for your comment. The double-J stent was inserted in the case of obstructive pyelonephritis or ureteral/renal calculi for which URS was planned (pre-stenting), and the stent placement was not used in the case when ESWL was performed in principle.
Page 2 line 43-46 in the revised manuscript
The double-J stent was inserted in the case of obstructive pyelonephritis or ureteral/renal calculi for which URS was planned (pre-stenting), and the stent placement was not used in the case when ESWL was performed in principle.
9)please specify according to which guidelines you decide when to perform SWL, ureteroscopy or PCNL
A9: Thank you for the comment. We basically followed the Japanese guideline on urolithiasis. We have described it as follows;
Page 2, line 34 in the revised manuscript
Based on Japanese guideline on urolithiasis[17]
10)When performing SWL, do you use sonography or fluoroscopy for localization of the stone?
A10: Thank you for the comment. When performing SWL, we used ultrasonography for localization of renal stone, whereas fluoroscopy was utilized for ureter stones. Given the radiolucent feature of the cystine stones, intravenous urography was utilized to support localization of the stone. We have revised the Method section as follows:
Page 2, line 44-46 in the revised manuscript;
When performing ESWL, we used ultrasonography for localization of renal stones, whereas fluoroscopy with or without intravenous urography was utilized for ureteral stones.
11)Please define exactly "a low likelihood of spontaneous passage."
A11: We agree. Based on the Japanese guideline on urolithiasis, we usually recommend intervention for stone which doesn’t move for one month. We have rephrased the sentence as follows:
Page 2, line 37-39 in the revised manuscript;
Patients were treated with URS for ureteral stones larger than 10 mm or stones that did not spontaneously pass for over one month.
Results
12)please specify what you understand by bilateral stone, kidney, kidney and ureter...
please reformulate "both stones"
A12: We agree. In this study, “bilateral stone” indicated the stones in both right and left side, and “both stones” indicated stones in the ureter and kidney. To clarify this point, we have revised the Result section as follows:
Page 3, line 12-14 in the revised manuscript;
At the initial presentation, 16 patients had stones in the bilateral side. Seventeen patients had only renal stones, whereas 3 patients had ureteral stones, and 2 patients had both renal and ureteral stones.
13)you mention 14 - staghorn calculi, but you performed 13 PCNL initially; 3 patients ureteral stones - 5 URS ???
Thank you for the question. In this study, 1 patient with staghorn calculi underwent URS (and ESWL). 1 patient with ureteral stones, 1 patient with renal stones and 2 patients with both renal and ureteral stones underwent URS. To make it clear, we have revised the manuscript as follows:
Page 3, line 17-21 in the revised manuscript;
13 patients with staghorn calculi underwent PCNL, 1 patient with staghorn calculi, 1 patient with ureteral stones, 1patient with renal stones and 2 patients with both renal and ureteral stones underwent URS (auxiliary ESWL was added at a physician’s discretion) and 2 patients with renal stones and 2 patients with ureteral stones underwent ESWL.
14)how do you evaluate the patient after first intervention?
Thank you for the question. After the first intervention, we evaluated via US 1 month after discharge. We have added the sentence to the Method section as follows;
Page 2, line 26 in the revised manuscript;
After the initial treatment, we evaluated via US 1 month after discharge.
15)16 patients - medical therapy (13/12/4) - what type of combination?
Thank you for the question. After diagnosis, sixteen patients took medical therapy, 3 were taking potassium citrate, 1 tiopronin, 1 sodium bicarbonate, 8 potassium citrate plus tiopronin, 2 tiopronin plus sodium bicarbonate and 2 potassium citrate plus tiopronin and sodium bicarbonate.
Page 3, line 23-26 in the revised manuscript;
After diagnosis, sixteen patients took medical therapy, 3 were taking potassium citrate, 1 tiopronin, 1 sodium bicarbonate, 8 potassium citrate and tiopronin, 1 tiopronin and sodium bicarbonate, and 2 were taking potassium citrate, tiopronin, and sodium bicarbonate
16)Figure 3 - please try to improve the quality and text (bold text on X- and Y-axis)
Thank you for the comment. We have improved the quality of the Figure 2 and 3.
Discussion
17)I think it is - In this study, we comprehensively presented "This result suggested the prophylactic role of SWL for asymptomatic cystine renal stones" - is interesting. The authors should try to elaborate more on this aspect. This can be one of the major conclusions of the manuscript
Thank you for the suggestion. We actively treated growing renal stones to avoid more invasive procedures (PCNL or open surgery). Interestingly, the usage of ESWL was high but incidence of surgical intervention except for ESWL was lower than other study. There have been several studies to distinguish ESWL-candidate or improve efficacy of ESWL by tiopronin. We have summarized this point and incorporated them into the revised manuscript as follows:
Page 8, line 20-29 in the revised manuscript
It has been reported that The CT feature of the stones may be helpful for candidate selection of ESWL [22, 23]. Moreover, tiopronin has been suggested to make cystine stone more fragile [25, 26]. Thus, early detection of “rough” cystine stones during regularly follow-up and active stone fragmentation via ESWL with CT and tiopronin could be a new approach to manage patients. Recently, URS is increasingly being used for the treatment of renal stones. Compared to ESWL, URS have an advantage in clearance rate of residual fragments, particularly for the lower pole stones [27]. Actually, patients who underwent flexible ureteroscopy in this study had fewer subsequent stone events and surgical interventions after that. The best timing and modality to treat growing renal stones should be further investigated.
Page X, line Y (the Conclusion section) in the revised manuscript:
This study suggested the potential of ESWL for prophylactic role.
We believe that incorporating your advice into the R1 version has improved the manuscript. Thank you once again.

Round 2
Reviewer 2 Report
This reviewers has no further comments.
Author Response
Response to Reviewer 2 Comments
This reviewers has no further comments.
Thank you very much for taking so many hours to review. We believe that incorporating your advice into the R1 version has improved the manuscript. Thank you once again.

Reviewer 3 Report
Dear Authors,
First of all, I want to congratulate you on your work. The manuscript is significantly improved.
Minor aspects:
Reference 17 is not appropriate, it is from Feb 2019, and your study finished in May 2019; and it does not represent the Japanese Guidelines.
Author Response
Response to Reviewer 3 Comments
Dear Authors,
First of all, I want to congratulate you on your work. The manuscript is significantly improved.
Thank you for your encouraging comment on our study. We agree with your suggestions and have incorporated our responses into the R2 version (Green color).
Minor aspects:
Q1:Reference 17 is not appropriate, it is from Feb 2019, and your study finished in May 2019; and it does not represent the Japanese Guidelines.
A1: Thank you for your point. We will replace the references with the latest and appropriate guidelines in japan.
Page11 line 21, in the revised manuscript:
- Kazumi T, Sung YC, Anthony CN, Manint U, Yung-Khan T, Yao LD; et al. The Urological Association of Asia clinical guideline for urinary stone disease. Int J Urol 2019, 26, 688-709
We believe that incorporating your advice into the R2 version has improved the manuscript. Thank you once again.
